SPECIAL ISSUE
THE EXTRACELLULAR ENVIRONMENT

# Cell-autonomous control coupled with tissue context regulates the cessation of migration at the site of organ development

Katsiaryna Tarbashevich[1,*], Zahra Labbaf[1,*], Moritz Ophaus[1], Jan Schick[1], Lucas Kühl[1], Sargon Gross-Thebing[1], Michal Reichman-Fried[1], Dennis Hoffmann[1], Martin Stehling[2], Jochen Seggewiss[3], Christian Ruckert[3], Johanna B. Kroll[4], Jan Philipp Junker[4,5] and Erez Raz[1,2,‡]

## ABSTRACT

Organ development relies on interactions among different cell types that form three-dimensional structures to carry out specific tasks. This process often involves active migration of progenitor cells toward specific positions within the embryo, where the cells then become immotile and form stable connections among themselves and with neighboring cells. In this work, we study the process of motility loss using zebrafish primordial germ cells (PGC) as an *in vivo* model. We show that changes in embryonic tissues as well as cell-autonomous events regulate the behavior of germ cells as they arrive at their target region. Importantly, we find that reduction in germ cell motility is correlated with the decay of RNA encoding for Dead end 1 (Dnd1), a conserved vertebrate RNA-binding protein that is essential for PGC migration. Indeed, decreasing or increasing the level of Dnd1 results in a premature or delayed stop to motility, respectively. These findings represent an RNA decay-based mechanism for timing the duration of cell migration *in vivo*.

KEY WORDS: Zebrafish, Primordial germ cell, Cell migration, Cell polarity, Gonad, Organogenesis, Dnd1

## INTRODUCTION

During organogenesis, progenitor cells often migrate toward specific positions within the embryo, where they stop and form stable connections with each other and with other cells and differentiate. Cessation of migration is controlled by mechanisms such as contact inhibition of locomotion or activation of specific receptors that provide a stop signal (e.g. for migrating T-cells) (Molon et al., 2022; Roycroft and Mayor, 2016; Shellard and Mayor, 2019). In addition, the motility can be regulated by desensitization of the chemoattractant receptor

[1]Institute of Cell Biology, ZMBE, Von-Esmarch-Straße 56, 48149 Muenster, Germany. [2]Max-Planck-Institute for Molecular Biomedicine, Roentgenstraße 20, 48149 Muenster, Germany. [3]Center for Medical Genetics, Clinic for Medical Genetics, University Hospital Muenster (UKM), University Muenster, Vesaliusweg 12-14, 48149 Muenster, Germany. [4]Quantitative Developmental Biology, Max-Delbruck-Centrum for Molecular Medicine (MDC), Hannoversche Str. 28, 10115 Berlin, Germany. [5]Institute of Pathology, Charite – Universitatsmedizin Berlin, Chariteplatz 1, 10117 Berlin, Germany.
*These authors contributed equally to this work

‡Author for correspondence (erez.raz@uni-muenster.de)

 E.R., 0000-0002-6347-3302

(Coombs et al., 2019; Kienle et al., 2021; Minina et al., 2007) and differential adhesion (Cortés et al., 2003), reviewed by Miskolci et al. (2021) and Yamaguchi and Knaut (2022).

A model for investigating the stop in migration in organogenesis *in vivo* is the gonad, because the progenitors of the germline, namely primordial germ cells (PGCs), often form away from the region where the gonad develops (Grimaldi and Raz, 2020). The chemokine Cxcl12 directs PGC migration, as was first demonstrated in zebrafish (Doitsidou et al., 2002; Knaut et al., 2003; Schick et al., 2025) and then in other vertebrates (Nishiumi et al., 2005; Stebler et al., 2004; Takeuchi et al., 2010; Herpin et al., 2008; Ara et al., 2003; Molyneaux et al., 2003), while in *Drosophila* the migration is directed by Hedgehog (Deshpande et al., 2025). Zebrafish PGCs arrive at the gonad region, maintain motility, interact with somatic cells and structures in the area (Paksa et al., 2016), and eventually stop. In this work, we study the mechanisms of motility loss in PGCs during early gonadogenesis. We show that the maternally provided RNA encoding for the Dead end1 (Dnd1) protein functions as a 'molecular clock' that controls the timing at which cell behavior switches from a motile to an immotile phase, allowing PGCs to form stable connections to somatic cells and generate gametes.

## RESULTS AND DISCUSSION

### Germ cell migration speed is reduced during development

To investigate the events occurring after PGCs reach the gonad region, we characterized cell behavior at 24 h post fertilization (hpf), when most of them have reached the target area (Weidinger et al., 1999), as well as 12 h later (36 hpf). At 24 hpf, the migration speed is about half of that observed for PGCs at 7-9 hpf (2 µm/min at the early stages; Reichman-Fried et al., 2004); 12 h later, it is lowered to about a quarter of its speed at 7-9 hpf (Fig. 1A,B). We also observed a strong decrease in displacement over 60 min (Fig. S3B), which was reduced from 12 µm at 24-26 hpf to 4 µm at 34-36 hpf (Fig. 1A,C; Movie 1). Thus, by 36 hpf PGCs exhibited little movement at the gonad region. The behavior of PGC at the target site can be controlled by cell-extrinsic factors, such as specific chemical or physical cues at the target region (Doren and Lehmann, 1997; Paksa et al., 2016), and potentially by cell-autonomous mechanisms.

To investigate the role of possible cues within the gonad region in the loss of PGC motility, we aimed to identify relevant genes expressed in somatic cells adjacent to PGCs at 24 hpf. To this end, we performed RNA tomography (tomo-seq) experiments (Junker et al., 2014). We sectioned embryos along the anterior-posterior or the dorsal-ventral axes and performed RNA sequencing (RNA-seq) on the individual tissue slices (Fig. S1A). This procedure provided us with a spatial RNA expression map of the gonad region.

Indeed, we could detect germ cell-expressed RNA molecules such as *ca15b* (Tarbashevich et al., 2015; Wang et al., 2013), *dnd1*

**DEVELOPMENT**

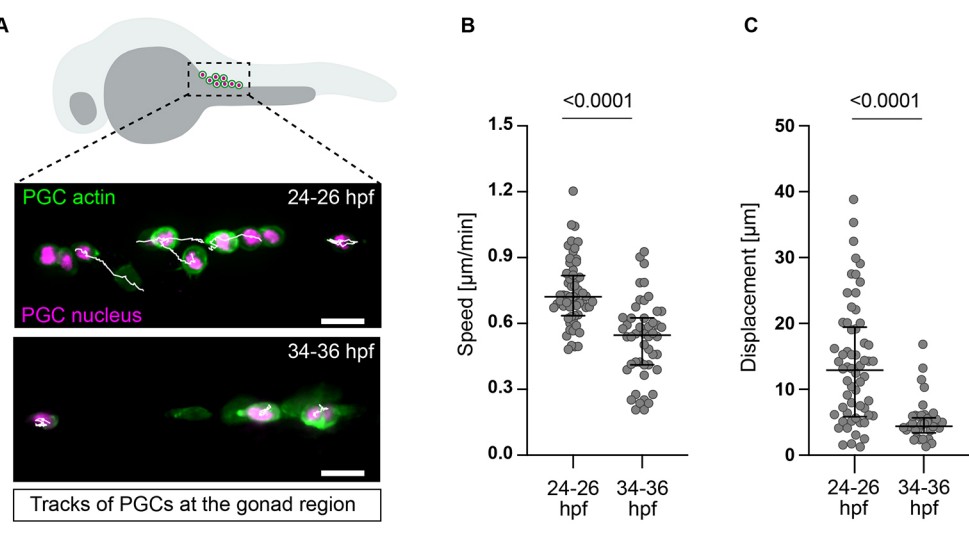

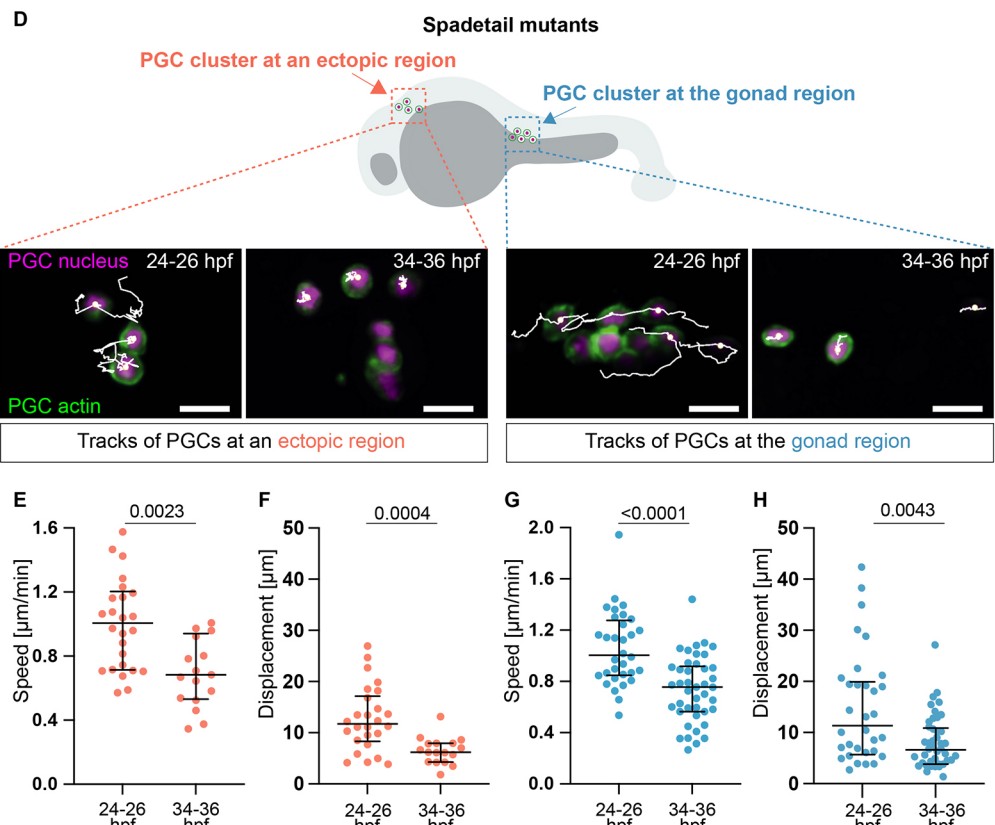

**Fig. 1. Reduction in PGC migration speed during development.** (A) PGCs at the gonad region at 24-36 hpf. (B,C) PGC speed (B) and displacement (C) over 60 min (24-26 hpf: 8 embryos, 60 PGCs; 34-36 hpf: 12 embryos, 48 PGCs). (D) PGC clusters at the gonad and the head at 24-36 hpf. (E,F) PGC speed (E) and displacement (F) in ectopic clusters over 60 min (24-26 hpf: 11 embryos, 26 PGCs; 34-36 hpf: 10 embryos, 17 PGCs). (G,H) PGC speed (G) and displacement (H) at the gonad over 60 min (24-26 hpf: 12 embryos, 32 PGCs; 34-36 hpf: 11 embryos, 42 PGCs). Unpaired two-tailed $t$-test. Data are median ± error bars (IQR). Scale bars: 20 µm.

(Weidinger et al., 2003) and *nanos3* (Köprunner et al., 2001). To identify genes with expression patterns similar to known PGC markers, we compared the spatial expression profiles to the germ cell marker *ddx4* (*vasa*). This analysis revealed that many genes expressed in the proximal convoluted tubule (PCT) segment of the pronephric duct (Wingert and Davidson, 2008) exhibited profiles similar to that of *ddx4* (Fig. S1B; Table S1). *In situ* hybridization experiments revealed that the PGCs were located ventral to the PCT (Fig. S2A,B).

To determine whether the interaction between PGCs and the PCT could affect PGC positioning, we ablated the PCT by expressing the bacterial toxin Kid within it, employing *UAS:kid* fish [*Tg(UAS:kid)*] (Labbaf et al., 2022) and fish expressing GAL4 and GFP in the PCT

[*Tg(PCT:GAL4FF;UAS:GFP)*; Kawakami et al., 2010]. Although Kid expression in the PCT depleted cells in this segment, as judged by GFP and *cdh17* mRNA expression (Fig. S2C-E), PGC cluster length and position as well as the PGC speed remained unaffected (Fig. S2F, right graph in G), suggesting that PGC positioning along the anterior-posterior axis of the embryo is not dictated by the PCT.

We have previously reported on the control of dorsoventral positioning of zebrafish PGC clusters by Wunen-family proteins expressed within the developing somites (Paksa et al., 2016). The localization of PGC clusters ventral to the PCT prompted us to investigate whether it can provide an additional level of regulation to PGC positioning along this axis. Indeed, elimination of the PCT

resulted in the shift of PGC clusters dorsally relative to the ventral somite border (Fig. S2C, schematic and left graph in G).

### Germ cell motility is reduced regardless of positioning within the embryo and clustering

To determine whether PGC migration speed reduction is a response to a specific cue or to conditions at the target site, we examined the behavior of PGCs in other locations at 24-26 hpf. Here, we examined the behavior of PGCs within an ectopic cell cluster in embryos mutated for the *spadetail* (*tbx16*) gene (Weidinger et al., 1999). Interestingly, ectopic PGCs exhibited a reduction in migration speed and displacement similar to that observed for PGCs following arrival at the gonad region (Fig. 1D-H), culminating in a loss of motility by 36 hpf. Thus, these migration parameters are reduced irrespective of PGC location.

To investigate whether the reduction of motility observed in cell clusters is related to enhanced interactions among the cells in the cluster or to specific interactions at the target region, we examined the behavior of single cells dispersed throughout the embryo lacking the chemokine receptor Cxcr4b (Doitsidou et al., 2002) (Fig. S3A). Interestingly, in this experiment too, we observed a reduced migration speed and displacement for single PGCs irrespective of their position (Fig. S3B-G).

### Maturation of the embryonic tissue is correlated with the reduction in migration speed

To examine whether the reduction in motility reflects global stage-related changes in the tissues within which PGCs migrate, we conducted transplantation experiments. Here, we transferred PGCs between embryos of different developmental stages and monitored their migration. To validate this experimental setup we first transferred cells between embryos of the same developmental stage (Fig. 2A,B). We analyzed the migration of both donor (green) and host (magenta) PGCs and found similar migration speeds between PGCs of the host and donor embryos (Fig. 2F). We then transplanted PGCs from 35 hpf embryos ("old" PGCs) into 4.5 hpf ("young") host embryos and followed their migration (Fig. 2C-E). Interestingly, the migration speed of the "old" PGCs transplanted into the "young" embryos was higher than that of the "old" PGCs located at the gonad region (Fig. S2C-E,G).

As embryonic development progresses, the expression of extracellular matrix (ECM) components in the embryo (e.g. fibronectin, collagen, etc.) increases (Gistelinck et al., 2016; Latimer and Jessen, 2010), consistent with the idea that, similar to other ameboid cell types (e.g. neutrophils, T cells, tumor cells), PGC migration could be hindered by the increasingly dense protein network of the tissue (Harmansa et al., 2023; Reis-Rodrigues et al., 2025; Wolf et al., 2013). To examine this possibility, we monitored the migration speed of PGCs expressing a combination of three membrane metalloproteases (MMPs), namely MMP2, MMP9 and MMP14, which have been shown to promote cell migration (Gonzalez-Molina et al., 2019; Orgaz et al., 2014). Notably, we observed an increase in the migration speed of such PGCs, consistent with the idea that PGC migration speed slows down as the embryo develops a denser network of ECM (Fig. 2H). Similar modulation of MMP activity and, thus, ECM remodeling is crucial for immune cells to switch their migration modes (e.g. macrophages; Travnickova et al., 2021) and for cancer cell epithelial-to-mesenchymal transition (Brassart-Pasco et al., 2020; Yamada et al., 2019). Together, these findings suggest that the maturation of the embryonic environment (e.g. ECM production and assembly) affects PGC migratory behavior.

To determine whether the environment alone can account for PGC decline in migration speed, we compared the behavior of 'old' transplanted PGCs to that of endogenous 'young' PGCs in 8 hpf host embryos (Fig. 2I). Surprisingly, while 'old' PGCs migrated faster within the 'young' embryo environment relative to their counterparts within 'old' sibling embryos (Fig. 2G), their migration speed was lower than that of 'young' host PGCs (Fig. 2I). Thus, the cessation of PGC motility at the target results from a combination of environmental effects and a PGC-intrinsic program.

### Cellular events correlated with the loss of motility

Next, we examined the cellular activities, distribution of polarity marker molecules and morphological features associated with PGC motility. Similar to other cell types that form blebs during migration (García-Arcos et al., 2024; Georgantzoglou et al., 2022; Paluch and Raz, 2013; Robertson et al., 2025; Schick and Raz, 2022; Schick et al., 2025), zebrafish PGCs form hydrostatic pressure-powered protrusions at the cell front during their migration (Blaser et al., 2006; Goudarzi et al., 2017; Truszkowski et al., 2023). Importantly, PGCs that form a reduced number of blebs show lower motility and impaired arrival at the target region (Blaser et al., 2006; Goudarzi et al., 2017). Thus, we quantified the frequency of bleb formation of PGCs at 24-26 hpf and 34-36 hpf (Fig. 3A,B,D). To eliminate possible effects of crowding at the clustering point and to reliably evaluate blebbing activity, we analyzed PGCs that were dispersed throughout the embryo (Fig. 3A). Indeed, we found that bleb frequency was significantly reduced from 24 to 36 hpf at any location within the embryo (Fig. 3B,D).

An additional process linked to the cell polarity is the accumulation of Actin at the cell front, while proteins like Ezrin accumulate at the cell back (Charras and Paluch, 2008; Hoffmann et al., 2025; Olguin-Olguin et al., 2021; Truszkowski et al., 2023). Thus, we analyzed the localization and distribution of markers for the cell front (Lifeact, green in representative images Fig. 3A, quantification in Fig. 3C,E) and back (Ezrin, magenta in representative images Fig. 3A) in PGCs located at the gonad and ectopic positions. This analysis revealed that at 24-26 hpf, PGCs exhibited distinct front-back polarity regardless of their location. In contrast, and in line with the loss of motility, by 34-36 hpf their polarity was lost, and Ezrin and Actin were more evenly distributed within the PGCs.

### Altered expression of proteins important for contractility and PGC identity

To determine the molecular basis for the reduction of blebbing activity (Fig. 3B,D) and to follow the differentiation state of the PGCs, the cells were isolated from 15, 25 and 35 hpf embryos. RNA-seq analysis was performed for PGCs at the gonad region (Fig. 3F) and for ectopically located germ cells (in embryos lacking functional guidance receptor Cxcr4b; Fig. S4A). This analysis revealed that between 15 and 35 hpf, mRNA levels of PGC markers such as *ddx4*, *nanos3*, *trd7a* and *dnd1* were reduced (Fig. 3G; Fig. S4B). Similarly, transcripts encoding proteins relevant to contractility and blebbing, such as Mylk and Rock2b (Amano et al., 1996; Blaser et al., 2006; Kishi et al., 1998; Totsukawa et al., 2004), were reduced (Fig. 3H; Fig. S4C). In contrast to a decrease in mRNA levels of specification and contractility markers, we observed an increase in differentiation- and adhesion-related transcripts (e.g. Fig. 3I; Fig. S4D). Similar transcriptomic profiles were observed in PGCs at the gonad region and in ectopic germ cells, consistent with the idea that PGC-intrinsic mechanisms control the loss of motility.

DEVELOPMENT

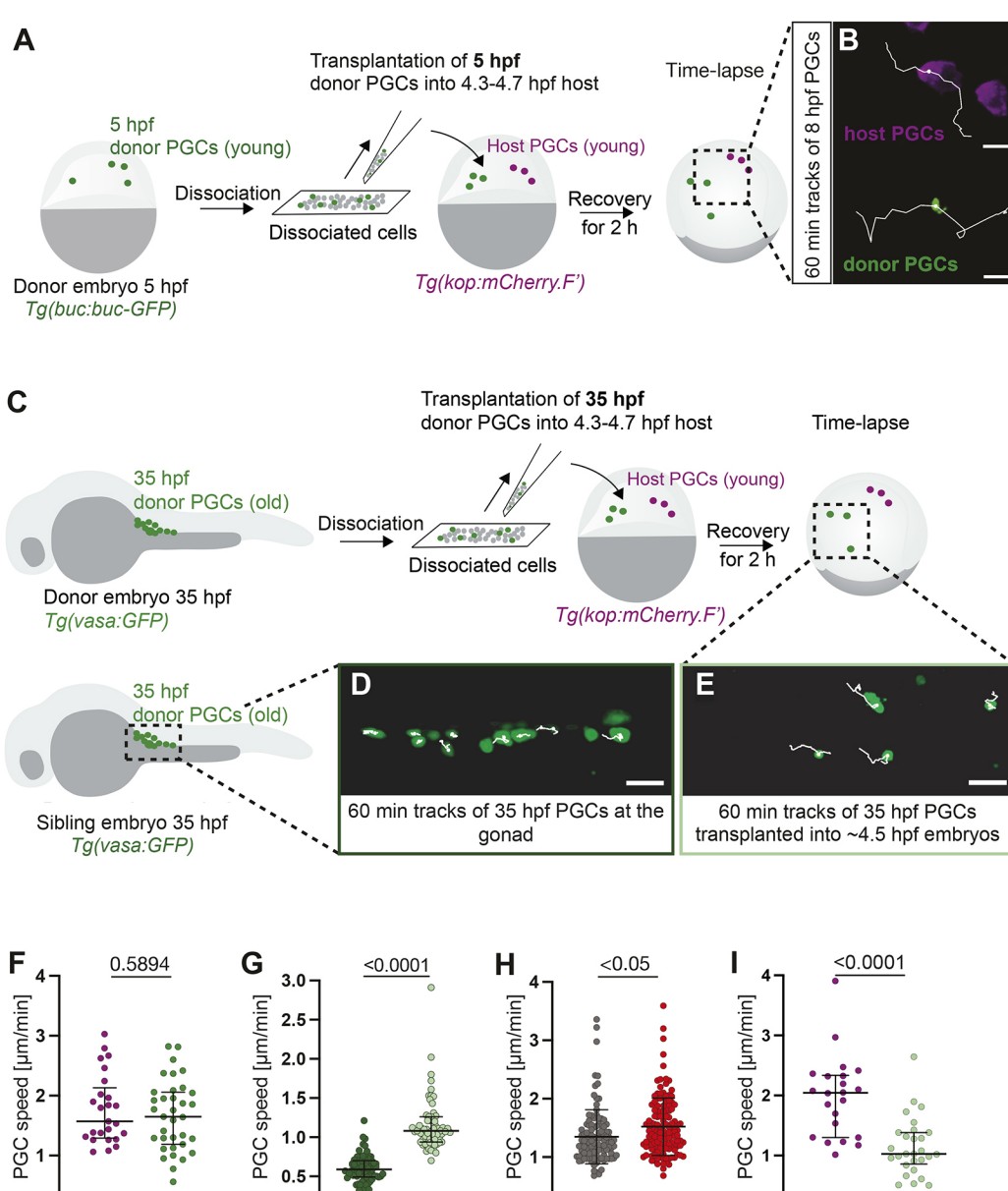

**Fig. 2. PGC migration speed is controlled by environmental cues and a cell-intrinsic program.** (A) Embryos at 5 hpf were dissociated, and the mixture of labeled PGCs (green) and somatic cells (gray) was transferred into wild-type embryos. (B) Tracks of 'young' donor PGCs and 'young' host PGCs within 'young' host (7-9 hpf). Scale bars: 20 µm. (C) Gonad regions of 35 hpf embryos were dissociated. This mixture of labeled PGCs (green) and somatic cells (gray) was transferred into wild-type embryos. (D,E) Tracks of 'old' PGCs in control embryos (no transplantation) at 35 hpf (D) and of 'old' donor PGCs transplanted into 'young' host (E). Scale bars: 30 µm. (F) Speed of 'young' host and donor cells within 'young' hosts (see B). 12 embryos with 25 PGCs (host); 20 embryos with 35 PGCs (donor). (G) Speed of 35 hpf ('old') PGCs at the gonad of control sibling embryos ('old') (see D) and within 'young' host embryos (see E). 19 embryos and 75 PGCs (35 hpf gonad), 14 embryos and 63 PGCs (35 hpf transplanted). (H) Speed of PGCs overexpressing MMPs, relative to control at 24-26 hpf. Control: 36 embryos, 110 PGCs; MMP overexpression: 36 embryos, 131 cells. (I) Speed of 'young' host (8 hpf) and 'old' donor (35 hpf) PGCs in 'young' host embryos (see C and E). 12 embryos, 22 PGCs (host); 28 PGCs (donor). Unpaired two-tailed $t$-test. Data are median±error bars (IQR) in F,G,I. In H, data are mean±s.d.

## *dnd1* expression level controls PGC motility

One way to explain the drop in cell motility is the dramatic decrease in the expression level of *dnd1* during the first 35 h of embryonic development. Indeed, we and others have previously found that Dnd1 is essential for germ cell motility and fate (Goudarzi et al., 2012; Gross-Thebing et al., 2017; Mall et al., 2021; Ruthig et al., 2019; Wang et al., 2025; Weidinger et al., 2003; Westerich et al., 2023; Youngren et al., 2005). The mechanism by which Dnd1 exerts its function involves binding target RNAs, stabilizing them and enhancing their translation (Aguero et al., 2017; Kedde et al., 2007; Ruthig et al., 2023; Zhang et al., 2021). Specifically, we previously found that Dnd1 function is required for enhancing

DEVELOPMENT

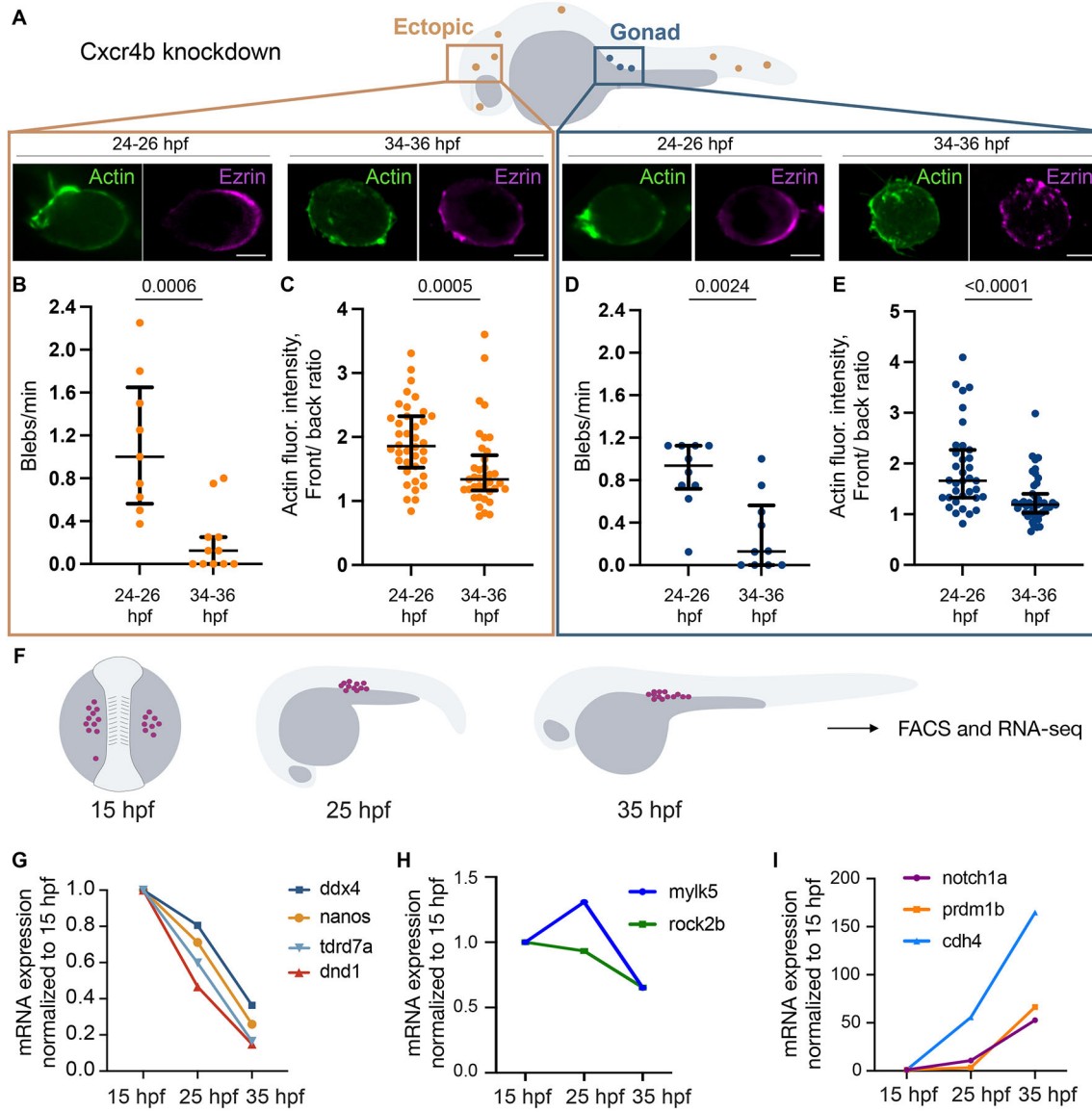

**Fig. 3. Cellular events during motility loss.** (A) PGCs at gonad region and ectopic positions were examined for blebbing activity and polarity. Localization of Lifeact and Ezrin in ectopic and gonadal PGCs. Scale bars: 5 μm. (B,D) Blebbing frequency in ectopic PGCs (B) and in PGCs at the gonad (D). Ectopic PGCs: 9 (24-26 hpf), 11 (34-36 hpf); gonadal PGCs: 10 (24-26 hpf), 10 (34-36 hpf). (C,E) Quantification of the PGC polarity based on the cell front/back ratio for the Lifeact fluorescence. Ectopic PGCs: 38 (24-26 hpf), 37 (34-36 hpf); gonadal PGCs: 35 (24-26 hpf), 43 (34-36 hpf). For B-E, Mann–Whitney test. Data are median±error bars (IQR). (F) Embryos at 15-35 hpf were used for FACS of PGCs (magenta) and sequencing. (G-I) Normalized levels of mRNAs important for PGC development (G), contractility (H) and adhesion/differentiation (I).

contractility and achieving Actin retrograde flow, while reducing the level of adhesion (Goudarzi et al., 2012). We, therefore, hypothesized that the progressive decay of maternally provided *dnd1* mRNA functions as a 'molecular clock' that dictates the rate of PGC motility loss.

To test this, we examined the effect of enhanced Dnd1 level on PGC behavior upon arrival at the gonad. Here, we injected embryos with a combination of RNAs encoding for germ plasm factors, to transform all the cells in the embryo into germ cells, hereafter referred to as 'induced PGCs' (iPGCs) (Wang et al., 2023). We further titrated down the amount of *dnd1* mRNA in the induction mix to the minimum sufficient to convert early blastomeres to PGCs (Dnd1-low), as determined by the successful arrival of such iPGCs at their target region by 24 hpf, as well as the expression of the PGC markers. To assess the effect of Dnd1 level on PGC migratory

behavior, we compared the Dnd1-low induction mix with one that included a very high level of *dnd1*-encoding RNA (Dnd1-high). iPGCs containing either of the two mixtures were transplanted into wild-type hosts lacking PGCs (Fig. 4A). We then analyzed iPGC migration and detected an increase in displacement and cell speed for iPGCs engineered to express more Dnd1 (Fig. 4B,C; Fig. S5A). We attribute the more pronounced differences in PGC displacement compared to the differences in speed to the fact that cells possessing higher contractility can better cope with obstacles in the environment without altering their migration direction as much. This allows PGCs to migrate further, which is manifested in the increase in displacement, while speed is less affected (Figs 1 and 4B,C). Furthermore, we performed transplantation experiments (Fig. 4A), employing wild-type fish as recipients (embryos with intact endogenous PGCs). The transplanted fish were subsequently

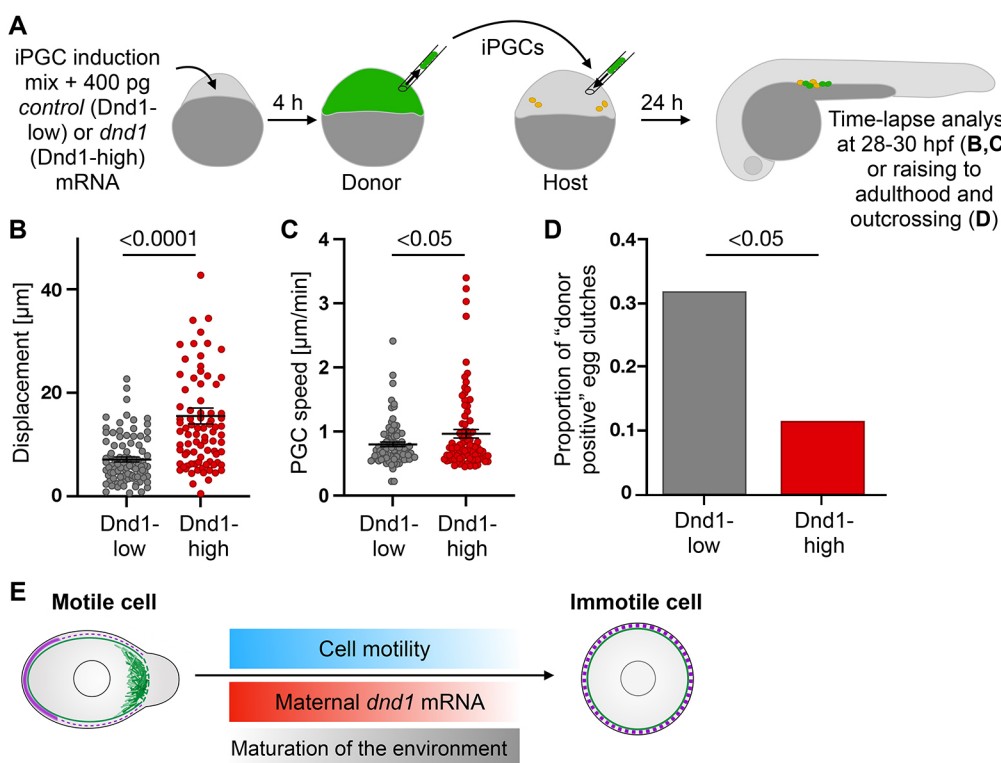

**Fig. 4. Control of motility loss by *dnd1*.** (A) iPGCs were transplanted either into PGC-depleted (experiments presented in B,C) or into wild-type (experiment presented in D and Fig. S5B) hosts. At 28 hpf, transplanted iPGCs were identified by the fluorescent marker (green). Endogenous PGCs in wild-type hosts are marked in orange. (B,C) iPGC displacement (B) and speed (C). Unpaired two-tailed *t*-test. Error bars: s.e.m. 91 (Dnd1-low) and 83 (Dnd1-high) PGCs. (D) iPGCs were transplanted into wild-type embryos that were raised and outcrossed. Fish were considered positive if they produced fluorescently labeled embryos. *P*=0.033, odds ratio=3.62 (Fisher's exact test); 41 (Dnd1-low) and 44 (Dnd1-high) animals. (E) Motile PGCs (left) express high levels of Dnd1 and are polarized [Actin in green at the cell front, and back markers at the rear (magenta)]. Later, *dnd1* level declines (red gradient), motility is reduced (blue gradient), while tissue rigidity (e.g. ECM level) and adhesion rise (gray gradient) culminating in stop of migration.

reared to adulthood, and the germline transmission efficiency of the genetically labeled donor cells was assessed. Notably, germ cells engineered to express higher Dnd1 levels (Dnd1-high) demonstrated a reduction in the proportion of offspring compared to Dnd1-low counterparts (Fig. 4D; Fig. S5B). We hypothesize that the reduced germline transmission rate of Dnd1-high PGCs can be attributed to their elevated contractility and limited capacity to form functional interactions with the somatic part of the developing gonad. An additional factor that could link PGC displacement with the ability to generate gametes is the fact that the position of PGCs along the migratory route could elicit changes in pluripotency, and epigenetic reprogramming (Jaszczak et al., 2025). These findings are consistent with the idea that *dnd1*-encoding RNA constitutes a limiting factor required for germ cell migration. Accordingly, the level of maternally provided *dnd1* RNA gradually decreases, such that by 36 hpf the protein function is minor, which results in polarity loss and a stop in migration (Fig. 4E). Interestingly, *in vitro* studies in the mouse model employing the gonad culture system have shown that germ cell motility gives rise to fractures in germline cysts, highlighting the importance of halting cell migration (Levy et al., 2024). Similar mechanisms may operate in other migratory cells when the transcription profile changes as part of differentiation, thus allowing cells to modulate motile activity as they undergo differentiation processes (e.g. myocardial precursors; Staudt and Stainier, 2012). Analogous to our findings on the role of the RNA-binding protein Dnd1 in zebrafish, the regulation of RNA stability, localization and translation controls a wide variety of cellular processes in other organisms. For example, the function of RNA-binding proteins that control RNA decay rates is crucial for regulating signaling cascades in neurogenesis as well as for the fidelity of neuronal cell migration (La Fata et al., 2014; Messaoudi et al., 2024; Zhang et al., 2025).

Taken together, our results suggest that the combination of cell-autonomous mechanisms, especially the decay rate of *dnd1* RNA,

and non-cell-autonomous mechanisms, such as ECM production, can control the timing at which cell migration stops, thereby allowing cells to interact with other cell types and form a functional organ (Fig. 4E).

## MATERIALS AND METHODS
### Zebrafish maintenance
Embryos were raised at 28°C in 0.3× Danieau's solution. The embryonic developmental stages were determined according to Kimmel et al. (1995). All the experiments and fish maintenance were performed following the regulations of the North Rhine-Westphalian Office of Nature, Environment and Consumer Protection (LANUV NRW). The list of transgenic and mutant animals employed in this work is presented in Table S2.

### DNA constructs, mRNA synthesis and microinjection
DNA constructs used in this work are listed in Table S3. Capped sense mRNAs were synthesized using the mMessage mMachine kit (Thermo Fisher Scientific) according to the manufacturer's instructions and kept at −20°C. Then 2 nl of an mRNA/morpholino sample was injected at the one-cell stage using glass capillaries and the PV830 Pneumatic PicoPump microinjector (WPI). Cxcr4b knockdown was achieved by injection of Cxcr4b morpholino oligonucleotide (Gene Tools, 5′-AAATGATGCTA-TCGTAAAATTCCAT-3′, 600 µM). Depletion of endogenous PGCs in transplantation experiments was achieved by injection of Dnd1 morpholino oligonucleotide (Gene Tools, 5′-GCTGGGCATCCATGTCTCCGACCAT-3′, 40 µM).

### Microscopy and acquisition analysis
Embryos were dechorionated in 0.3× Danieau's solution [6-8 min at room temperature (RT)]. For embryos older than 18-19 hpf, 200 mg/l of tricaine (Sigma-Aldrich) was added before imaging to inhibit movement. To prevent pigmentation, embryos after 20 hpf were transferred to 0.003% of 1-phenyl 2-thiourea (PTU) (Sigma-Aldrich) in 0.3× Danieau's solution. Dechorionated embryos were transferred to agarose coated (Thermo Fisher Scientific) ramps with 0.3× Danieau's solution. Dechorionated embryos older than 24 hpf were mounted laterally in 0.5% LMP agarose (Invitrogen) drops that contained 200 mg/l of tricaine (Sigma-Aldrich). Polymerized

agarose drops were covered with 0.3× Danieau's solution that contained 200 mg/l of tricaine (Sigma-Aldrich). During the imaging, the embryos were maintained at 28°C using a heating stage (PECON). The embryos were subjected to image acquisition using 10×, 20× and 40× water immersion objectives and z-stacks were acquired from each sample on a spinning disk confocal microscope (Zeiss Imager M2, Visitron systems) using a Hamamatsu digital camera (C13440-20CU ORCA-flash 4.0) controlled by the VisiView software (version 4.5.0.14). The z-stacks were captured within a range of 6-30 µm with 2-4 min intervals (for a total of 60-120 min). PGC tracking, tissue drift correction and measurements of PGC speed and displacement were performed using Imaris software (version 9.9.0). The polarity of the migrating PGCs inside and outside the gonad region was investigated by following the localized distribution of Lifeact using *Tg(kop: lifeact.mCherry)* and Ezrin using *Tg(kop:YPET.Ezrin)* fishlines at 24 to 26 hpf and at 34 to 36 hpf. 14 µm Z-stacks were acquired using a 63× water immersion objective every 2 µm with 8 s frame intervals. The 4D data were analyzed with Fiji (version 2.1.0/1.53c) and Imaris software (version 9.7.2) allowing a detailed analysis of the localized distribution of polarity markers. The blebbing frequency of the migrating PGCs inside and outside the gonad region was analyzed by farnesylated mCherry (membrane labeling) using *Tg(kop:mCherry.F')* and Lifeact distribution using *Tg(kop:EGFP-lifeact)* fishlines at 24-26 hpf and at 34-36 hpf. We acquired 14 µm z-stacks using a 63× water-immersion objective every 2 µm with 5 s per frame intervals (for a total of 10 min). The 4D data were analyzed with the Fiji (version 2.1.0/1.53c). We considered 5-8 min of the acquired data for analyzing the number of blebs.

### Transplantation experiments

For experiments presented in Fig. 2, donor embryos were dechorionated and treated with trypsin (Sigma-Aldrich) for 5 min at 31°C followed by quick washes with dissociation buffer (Gibco). Next, the embryos were dissociated by slow pipetting with a 100 µl pipette in 50-80 µl of the cell dissociation buffer (Gibco). The dissociated cells were transferred to a glass slide and ∼50 cells were immediately transplanted into host embryos. For experiments presented in Fig. 4 and Fig. S5, donor iPGCs were induced as described by Hoffmann et al. (2025) with addition of 400 pg of either control (Dnd1-low) or *dnd1* (Dnd1-high) mRNA. At ∼4 hpf iPGCs from such donor embryos were transplanted into the hosts of the same age without previous dissociation. The transplantation was conducted using a stereomicroscope (Leica S8AP0) with a 50 µm transplantation needle with spike (BM100T-10, Biomedical Instruments). Afterwards the embryos developed at 28°C for 2 h followed by microscopy. For 35 hpf *Tg(vasa: vasa-EGFP)* donor embryos, the gonad region was micro-dissected before dissociation. Wild-type AB or *Tg(kop:mCherry.F')* embryos at 4.3-5 hpf were employed as hosts.

### PGC polarity analysis

To quantify Actin signal asymmetry, mean fluorescence intensity of the Lifeact was measured at the cell boundary using a circular region of interest (ROI) with a diameter of 45 pixels. The mean intensity was measured at the brightest area of the Lifeact signal. The second measurement was taken at the opposing aspect of the cell boundary using an ROI of identical dimensions. Asymmetry ratio was calculated by dividing the mean intensity at the brightest region by the mean intensity at the opposing region. Analysis was performed in ImageJ Version: 2.16.0/1.54p.

### RNA tomography (Tomo-seq)

This method was performed as described previously in Junker et al. (2014). In short, 24 hpf *Tg(kop:EGFP.F')* embryos were covered with Jung tissue freezing medium (Leica), oriented laterally, and the region of the developing gonad was marked by polyacrylamide beads. Next, embryos were frozen on dry ice. The labeled region (developing gonad) was sectioned into 60 sections (10 µm thickness) along the anteroposterior axis or into 48 sections (10 µm thickness) along the dorsoventral axis. Each section was transferred to pre-cooled LoBind 1.5 ml tubes (Eppendorf) and kept on dry ice. RNA from each section was extracted and subjected to RNA-seq (for details see the corresponding Supplementary Materials and Methods section).

### RNA-seq

PGCs were sorted by fluorescence activated cell sorting (FACS), mRNA was extracted and subjected to RNA-seq (for details see the corresponding Supplementary Materials and Methods section).

### Whole mount *in situ* hybridization and RNAscope

Preparation of the embryos and whole mount *in situ* hybridization was performed following the protocol described in (Thisse and Thisse, 2008). After staining, embryos were covered with 80% glycerol (Applichem) and positioned in the correct orientation. Color images were acquired using Leica MZ16F stereomicroscope with a AxioCam MRc5 camera controlled by ZEN software (version 2012 blue edition, 1.1.1.0).

RNAscope was performed by using the RNAscope Multiplex Fluorescent Kit (Bio-techne) following the protocol described in Gross-Thebing et al. (2014). The embryos were mounted laterally in 1% LMP agarose drops (Invitrogen) and covered with 1× PBS. Fluorescent z-stacks were acquired on a confocal microscope (LSM 710, Zeiss) controlled by the ZEN software (version 2010B SP1) using 20-40× water immersion objectives (Zeiss). The acquired images were processed using Fiji (version 2.1.0/1.53c) by generating maximum intensity z-projections of the 42 focal planes containing the fluorescence signal. Brightness and contrast were adjusted similarly in both the control and treatment samples for each channel. 3D-reconstructions from fluorescent images were made in Imaris (version 9.7.2).

### PGC induction

PGCs were induced as described in Hoffmann et al. (2025). In brief, one-cell-stage embryos were injected with the mix of six mRNAs (see Table S3): 100 pg *nanos3CDS.globin3′utr* (internal DB number: 278), 226 pg *vasaFL.globin3′utr* (333), 40 pg (Dnd1-low) or 400 pg (Dnd1-high) *dnd1.globin3′utr* (487), 300 pg *tdrd7.globin3′utr* (A241), 16 pg *pCS2-buc* (B639), 19 pg *tdrd6.globin3′utr* (F089), and 400 pg (Dnd1-low) or 40 pg (Dnd1-high) of stuffer mRNA. Fluorescent marker mRNA with the *nanos3* 3′ untranslated region (UTR) [e.g. *mGFP.nanos3′utr* (355)] was included to control for the successful PGC induction at 5-6 hpf.

### Statistics

Statistical analysis was performed using Prism software (version 9). The normality of the distribution of the data was tested with D'Agostino-Pearson omnibus normality test. The significance of the data was tested with either the unpaired two-tailed Student's *t*-test (if the data were normally distributed), by Mann–Whitney *U*-test (if the data were not normally distributed) or by Fisher's exact test. The *P*-values are shown on top of the graphs. If not stated otherwise, statistical analysis was performed on experiments with three independent biological replicates.

### Acknowledgements

We thank Ursula Jordan, Esther-Maria Messerschmidt and Ines Sandbote for excellent technical help. Z.L. was and L.K. is a member of CiM-IMPRS, the joint graduate school of the Cells-in-Motion Interfaculty Centre, University of Münster, Germany and the International Max Planck Research School – Molecular Biomedicine, Münster, Germany. We thank Celeste Brennecka for editing the manuscript. The flow cytometry experiments were performed at the flow cytometry unit of the Max-Planck-Institute for Molecular Biomedicine, and RNA-seq at the Clinic for Medical Genetics.

### Competing interests

The authors declare no competing or financial interests.

### Author contributions

Conceptualization: E.R., K.T., Z.L.; Funding acquisition: E.R.; Investigation: K.T., Z.L., J. Schick, L.K., S.G.-T., M.O., M.R.-F., D.H., J.P.J.; Methodology: K.T., Z.L., J. Schick, L.K., S.G.-T., M.O., M.R.-F., D.H., M.S., J. Seggewiss, J.P.J., C.R., J.B.K.; Project administration: E.R., K.T.; Resources: E.R.; Supervision: E.R., K.T.; Validation: K.T.; Visualization: K.T., Z.L.; Writing – original draft: E.R., K.T., Z.L.; Writing – review & editing: E.R., K.T., J. Schick, L.K., M.R.-F., D.H., M.S., J.P.J.

### Funding

M.O., J. Schick, D.H., L.K. and E.R. are supported by funds from the Westfälische Wilhelms-Universität Münster, the Max-Planck-Institut für Molekulare Biomedizin,

the Deutsche Forschungsgemeinschaft (grant RA863/14-1 and the SFB 1348 project B06). J.B.K. and J.P.J. are supported by the Deutsche Forschungsgemeinschaft project number 540370162. Open Access funding provided by University of Muenster. Deposited in PMC for immediate release.

**Data and resource availability**
Source data and plasmids generated in this study are available from E.R. Raw data of the RNA-seq experiments presented in the manuscript are available on GEO under accession numbers GSE310575 (bulk mRNA-seq) and GSE313050 (RNA tomography). All other relevant data and details of resources can be found within the article and its supplementary information.

**Peer review history**
The peer review history is available online at https://journals.biologists.com/dev/lookup/doi/10.1242/dev.205271.reviewer-comments.pdf

**Special Issue**
This article is part of the Special Issue 'The Extracellular Environment in Development, Regeneration and Stem Cells', edited by Alex Hughes and Rashmi Priya. See related articles at https://journals.biologists.com/dev/issue/153/16

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
