## [Peer Review File · Development (Cambridge, England)]

Cell-autonomous control coupled with tissue context regulates the cessation of migration at the site of organ development

Katsiaryna Tarbashevich, Zahra Labbaf, Moritz Ophaus, Jan Schick, Lucas Kühl, Sargon Gross-Thebing, Michal Reichman-Fried, Dennis Hoffmann, Martin Stehling, Jochen Seggewiss, Christian Ruckert, Johanna B. Kroll, Jan Philipp Junker and Erez Raz
DOI: 10.1242/dev.205271

Editor: James Turner

Review timeline

Original submission:	26 September 2025
Editorial decision:	13 November 2025
First revision received:	19 December 2025
Accepted:	16 January 2026

Original submission

First decision letter

MS ID#: dev.205271

MS TITLE: Cell-autonomous control coupled with tissue context regulates the cessation of migration at the site of organ development

AUTHORS: Katsiaryna Tarbashevich; Zahra Labbaf; Jan Schick; Lucas Kühl; Sargon Gross-Thebing; Moritz Ophaus; Michal Reichman-Fried; Dennis Hoffmann; Martin Stehling; Jochen Seggewiss; Jan Philipp Junker; Christian Ruckert; Johanna B. Kroll; Erez Raz

Dear Dr Raz,

I have now received all the referees' reports on the above manuscript, and have reached a decision. The referees' comments are appended below.

As you will see, the referees express considerable interest in your work, and many of the suggested revisions are minor. However, they also request provision of further data and clarifications, which would be important to resolve before we can consider publication. If you are able to revise the manuscript along the lines suggested, I will be happy to receive a revised version of the manuscript. Your revised paper will be re-reviewed by one or more of the original referees, and acceptance of your manuscript will depend on your addressing satisfactorily the reviewers' major concerns. Please also note that Development will normally permit only one round of major revision. If it would be helpful, you are welcome to contact us to discuss your revision in greater detail. Please send us a point-by-point response indicating your plans for addressing the referees' comments, and we will look over this and provide further guidance.

Please attend to all of the reviewers' comments and ensure that you upload both a 'clean' version of your Word file, along with a highlighted version clearly showing where you have made changes in the revised manuscript. Please avoid using 'Tracked changes' in Word files as these are lost in PDF conversion. I should be grateful if you would also provide a point-by-point response detailing how you have dealt with the points raised by the reviewers in the 'Response to Reviewers' box. If you do not agree with any of their criticisms or suggestions please explain clearly why this is so.

Reviewer 1

This work analyses the migration of progenitor germ cells (PGC) in zebrafish embryos, with particular attention to the mechanism that controls their cessation of migration. The manuscript reports that PGC cessation of migration depends on a combination of cell-autonomous and environmental factors. Specifically, they identify the expression of Dead End 1 (Dnd1) as playing an important role in stopping cell migration.

This is a focused and well-done manuscript that deals with the important and rather unexplored problem of how cells stop their migration once they reach their destination. The results are clear and convincing (especially the cell autonomous, more than the environmental factors). The author needs to address the following issues before publication in Development

1. The authors characterise the cessation of PGC migration by referring to speed and displacement (Fig. 1A-C). In the text, they refer to displacement per unit time ($1\frac{1}{4}\mu\text{m}/\text{h}$). Please clarify the difference between velocity and displacement per unit time; otherwise, the distinction seems redundant.
2. The authors explore the role of the proximal convoluted tubule (PCT) in PGC migration by genetically ablating the PCT and showing no major change in the distribution of cells, except for a shift in the dorso-ventral position. However, as the focus of this work is cell motility rather than cell distribution, the authors should examine whether PGC motility and its cessation are affected by PCT ablation.
3. Figure 2A-D describes an important graft experiment, but the labelling in the figure is somewhat confusing, particularly the X-axis in panel D.
4. From Figure 2, it is concluded that PGC motility responds to ECM changes occurring during embryo development. This is an interesting and logical idea; however, the evidence presented is rather weak. The graft experiments (Fig. 2A-D) clearly indicate that an environmental factor changing with developmental stage affects PGC motility. Nevertheless, there are many possible candidates for this factor. The observation that expressing metalloproteases in PGCs increases their speed does not demonstrate that the ECM—the substrate of metalloproteases—is the factor responsible for the environmental change controlling PGC motility.

Reviewer 2

SUMMARY OF THE ADVANCE MADE IN THIS PAPER AND ITS POTENTIAL SIGNIFICANCE TO THE FIELD

Here, Tarbashevich et al used migrating zebrafish Primordial Germ Cells (PGCs) as an in vivo model to understand how cells lose motility as they reach their target. Through live imaging and transplantation experiments, the authors demonstrate that migrating PGCs slow down, decrease migratory blebs, and lose polarity as development progresses independent of PGC proximity to the target gonad. While they provide some evidence that the extracellular matrix impacts PGC speed, their transplantation experiments suggest progressive loss of motility is due to autonomous changes in PGCs. After characterizing transcriptomic profiles of PGCs sorted from embryos at different developmental stages, they focused on the role Deadend1. They were able to titrate the amount of Deadend1 and found that increasing Deadend1 levels in transplanted PGCs increased their displacement at the cost of contributing to the next generation, presumably due to sustained motility.

Strengths:

- 1) The authors' clever use of mutants revealed that loss of motility is not simply due to proximity to their target, which represents a significant advance in our understanding of the regulation of motility in vivo.

- 2) The transplantation experiments provide strong evidence for an intrinsic motility clock in PGCs, shedding light on a long-standing question in the field of PGC development and migration.
- 3) Sample sizes for speed and displacement quantifications were large.
- 4) The manuscript is well-written with clear logic and significance.
- 5) For the most part, the claims are well-supported by the data, with a few exceptions listed below.

SUGGESTIONS TO AUTHORS

- 1) Unlike other perturbations in this manuscript, speed was not quantified when *Deadend1* was increased. Instead, germline transmission efficiency was measured as a proxy for PGC migration. The manuscript's central claim that *Deadend1* correlate with motility loss would be stronger with direct speed measurements.
- 2) Raw data in the form of demultiplexed .fastq files do not appear to have been provided, which prevents assessment of sequencing data quality. The counts tables provided were processed data that don't allow assessment of library quality (quality score distribution, percentage of unique and duplicated reads, GC bias, etc). Original data should be uploaded into a Sequence Read Archive with all metadata descriptors before publication.
- 3) Authors state in the text that front/back polarity is lost by 34-36 hpf but only images of four PGCs are shown. Quantification of front/back polarity would better support this claim.

Minor comments:

Text citing Figure 2D states that old PGCs migrate faster in a younger environment - but it's unclear what the younger environment is based on the figure and the figure legend.

7 hpf is discussed in the text but the data is not shown (Fig 1).

Reviewer 3

SUMMARY OF THE ADVANCE MADE IN THIS PAPER AND ITS POTENTIAL SIGNIFICANCE TO THE FIELD

This is an interesting paper investigating the mechanism of how cells stop migration during development. Previous literature has shown that the dynamic responses of receptors to external ligands are important in cell stopping. Here the authors introduce an intrinsically programmed mechanism to stop cells based on changes of expression of an RNA-binding protein, which in turn regulates migration-associated genes. These findings have broad relevance, as they may suggest mechanisms by which other cell types cease migration in various contexts, for example in disease pathologies, such as cancer metastasis. The experimental set includes genetic manipulations, transplantations and live imaging and generally supports the claims of the authors. The paper also includes elegant approaches such as RNA tomography and tissue-specific gene silencing to obtain spatial maps of gene expression and to interrogate relevant mechanisms. The data presentation and writing are of high quality, and I only have minor comments and suggestions for improvement.

SUGGESTIONS TO AUTHORS

Minor points:

-In the gene silencing section, Fig. S1C and D, it would be helpful to have a quantitative estimation of the level of knockdown, to indicate the levels achievable and relate to phenotypic changes observed in S1E and S1F.

-RNA tomography is mentioned at an early stage in the manuscript, but we don't see relevant data before figure S4. Although there is a table with data, it would be good to have a diagram of the assay and charts of the relevant datasets.

-The transplantation experiments are important to distinguish cell autonomous from non-cell autonomous effects. The controls included in figure S3 may be helpful to include in the same main figure, to facilitate interpretation. In addition, can the authors comment on the estimated purity of the transplants in PGCs. Is there a significant proportion of non-PGC cells carried over that may contribute to phenotypes?

-The actin/Ezrin distribution is suggested to be altered during PGC development, but this would need some quantification.

-In the cartoon diagram of Figs 3F and S4 can the authors use consistent references to 'RNA-seq' versus 'NGS'.

-In Fig. S4, the authors report transcriptome data from Cxcr4b knockout fish but the rationale of the use of this strain is not clearly explained in the results.

-In this sentence, can the authors give the names of all the RNAs injected "we injected embryos with a combination of RNAs encoding for germ plasm factors, including Dnd1, to transform all the cells in the embryo into germ cells, hereafter referred to as "induced PGCs" (iPGCs) (Wang et al., 2023)."

-Can the authors more clearly comment on a possible relevance for the observation that dnd-high cells have lower germ-line transmission? Could this suggest that there is a functional/developmental trade off with PGC migration? It would be good to discuss this point along with other references where such trade-offs might have been observed.

-The authors could give some more background into the biological function of dnd, to better appreciate the biological mechanisms and cite any other studies where similar regulation of migration might have been observed.

-In the introduction, it would be helpful to give more background on alternative mechanisms of stopping, for example chemoattractant receptor desensitization (e.g. Minina et al. 2007, Coombs et al. 2019, Kienle et al. 2020), and differential adhesion (Yamaguchi et al. 2022). This will help appreciate better the novelty of this paper regarding cell-intrinsic mechanisms of stopping.

-Another suggestion for the last paragraph of the paper is to discuss the broader relevance of the findings, in other cell types and biological contexts.

First revision

Author response to reviewers' comments

We thank the reviewers for their work on evaluating our manuscript and for the very constructive comments that improved the quality of the paper. Our responses to individual reviewers' comments are provided below as answers to the specific issues raised. We hope that the reviewers find this version of the paper suitable for publication as a Report in Development. As a general note, we considered the scope and volume of the results to fit the "Report" format of Development, rather than a full "Research article". This format is much more limited in the number of figures and text length. Thus, some of the terms and issues were indeed not always fully explained. We have made an effort to improve on this point, but in some cases we could only refer the readers to other publications.

Comments from the Reviewers:

Reviewer 1: This work analyses the migration of progenitor germ cells (PGC) in zebrafish embryos, with particular attention to the mechanism that controls their cessation of migration. The manuscript reports that PGC cessation of migration depends on a combination of cell-autonomous and environmental factors. Specifically, they identify the expression of Dead End 1 (Dnd1) as playing an important role in stopping cell migration.

This is a focused and well-done manuscript that deals with the important and rather unexplored problem of how cells stop their migration once they reach their destination. The results are clear and convincing (especially the cell autonomous, more than the environmental factors). The author needs to address the following issues before publication in Development

1. The authors characterise the cessation of PGC migration by referring to speed and displacement (Fig. 1A-C). In the text, they refer to displacement per unit time ($\mu\text{m}/\text{h}$). Please clarify the difference between velocity and displacement per unit time; otherwise, the distinction seems redundant.

We thank the referee for noticing this point, which was not clear. We now provide a scheme that explains the way we measured speed and displacement (new Fig. S3B) and rephrased the sentence as follows: “We also observed a strong decrease in displacement, which was reduced from 12 μm at 24-26 hpf to 4 μm at 34-36 hpf per 1 hour of PGC migration.”

2. The authors explore the role of the proximal convoluted tubule (PCT) in PGC migration by genetically ablating the PCT and showing no major change in the distribution of cells, except for a shift in the dorso-ventral position. However, as the focus of this work is cell motility rather than cell distribution, the authors should examine whether PGC motility and its cessation are affected by PCT ablation.

As requested by the referee, we analyzed PGC migration speed in the presence (control) and absence of the PCT segment. The result (no difference in speed) is now added to the new Fig. S2 (Fig. S2G, right graph).

3. Figure 2A-D describes an important graft experiment, but the labelling in the figure is somewhat confusing, particularly the X-axis in panel D.

We apologize for the confusing labelling. We have now changed the text of the figure legend and the labeling of the X-axis. The Figure was also modified according to the request of the other referee (referee 3) such that the former panel D is now designated as G in the new version of Figure 2.

4. From Figure 2, it is concluded that PGC motility responds to ECM changes occurring during embryo development. This is an interesting and logical idea; however, the evidence presented is rather weak. The graft experiments (Fig. 2A-D) clearly indicate that an environmental factor changing with developmental stage affects PGC motility. Nevertheless, there are many possible candidates for this factor. The observation that expressing metalloproteases in PGCs increases their speed does not demonstrate that the ECM—the substrate of metalloproteases—is the factor responsible for the environmental change controlling PGC motility.

Indeed, our experimental setup does not exclude other possibilities of environmental influence beyond the ECM. In response to this point, we changed the text such that it puts into focus the environment in general and not only on the ECM aspect: “Together, these findings suggest that the maturation of the embryonic environment (e.g., ECM production and assembly) affects PGCs’ migratory behavior.”

Additionally, we changed “ECM” in the final model (Fig. 4E) to “Maturation of the environment “.

Reviewer 2: SUMMARY OF THE ADVANCE MADE IN THIS PAPER AND ITS POTENTIAL SIGNIFICANCE TO THE FIELD

Here, Tarbashevich et al used migrating zebrafish Primordial Germ Cells (PGCs) as an in vivo model to understand how cells lose motility as they reach their target. Through live imaging and transplantation experiments, the authors demonstrate that migrating PGCs slow down, decrease migratory blebs, and lose polarity as development progresses independent of PGC proximity to the target gonad. While they provide some evidence that the extracellular matrix impacts PGC speed, their transplantation experiments suggest progressive loss of motility is due to autonomous changes in PGCs. After characterizing transcriptomic profiles of PGCs sorted from embryos at different developmental stages, they focused on the role Deadend1. They were able to titrate the amount of Deadend1 and found that increasing Deadend1 levels in transplanted PGCs increased their displacement at the cost of contributing to the next generation, presumably due to sustained motility.

Strengths:

- 1) The authors' clever use of mutants revealed that loss of motility is not simply due to proximity to their target, which represents a significant advance in our understanding of the regulation of motility in vivo.
- 2) The transplantation experiments provide strong evidence for an intrinsic motility clock in PGCs, shedding light on a long-standing question in the field of PGC development and migration.
- 3) Sample sizes for speed and displacement quantifications were large.
- 4) The manuscript is well-written with clear logic and significance.
- 5) For the most part, the claims are well-supported by the data, with a few exceptions listed below.

SUGGESTIONS TO AUTHORS

1) Unlike other perturbations in this manuscript, speed was not quantified when Deadend1 was increased. Instead, germline transmission efficiency was measured as a proxy for PGC migration. The manuscript's central claim that Deadend1 correlate with motility loss would be stronger with direct speed measurements.

We thank the referee for pointing at this issue, which we try to clarify in the figures and the text.

In Figure 4, we observed an increase in displacement of cells in which Dnd1 expression is elevated (Fig 4B). We consider the germline transmission efficiency presented in Fig 4D to be an outcome of the increased displacement.

Following the referee's suggestion, we have now included PGC speed measurements at Dnd1-low and Dnd1-high conditions. The new panel with the quantification is now presented in Fig. 4C. In general, in our paper, differences in displacement are more pronounced than the effects on speed (e.g., in Fig 1 as well). An increase in contractility can indeed potentially increase both parameters. Thus, the effect of contractility and polarity on cell migration speed could be limited by the environment more than the effect on the directional persistence of the migration. We refer to this point in the manuscript as - "We attribute the more pronounced differences in PGC displacement compared to the differences in speed to the fact that cells possessing higher contractility can better cope with obstacles in the environment without altering their migration direction as much. This allows PGCs to migrate further, which is manifested in the increase in displacement, while speed is less affected (Fig. 1, Fig. 4B, C)."

2) Raw data in the form of demultiplexed .fastq files do not appear to have been provided, which prevents assessment of sequencing data quality. The counts tables provided were processed data that don't allow assessment of library quality (quality score distribution, percentage of unique and duplicated reads, GC bias, etc). Original data should be uploaded into a Sequence Read Archive with all metadata descriptors before publication.

The raw data of all RNA-seq experiments presented in the manuscript have now been submitted to GEO. The GEO accession numbers: GSE310575 (bulk sequencing, open access) and GSE313050 (RNA tomography data, <https://www.ncbi.nlm.nih.gov/geo/query/acc.cgi?acc=GSE313050>, Reviewer token: cfybsegwzjwprgt). The accession numbers are also included in the text of the manuscript.

As an additional control for the quality of the RNA-seq data, we include in the new version of the paper graphs with examples of RNA molecules encoding for proteins that play roles in adhesion and differentiation. Here, we observe an increase in expression level between 15hpf and 36hpf. This shows that the decrease presented in the original version of the paper is specific and is not a result of a global reduction in expression among the time points (new Figs. 3I and S4D).

3) Authors state in the text that front/back polarity is lost by 34-36 hpf but only images of four PGCs are shown. Quantification of front/back polarity would better support this claim.

In response to this point, we conducted additional experiments and analyzed a larger number of cells. The quantification was performed by measuring front-to-back ratios of the fluorescent actin signal. The results are now presented in the Fig. 3C, E.

Minor comments:

Text citing Figure 2D states that old PGCs migrate faster in a younger environment - but it's unclear what the younger environment is based on the figure and the figure legend.

We have changed the labeling in the Figure (Figure 2G in the current version) and the Figure legend accordingly, making the host and donor labeling as clear as possible.

7 hpf is discussed in the text but the data is not shown (Fig 1).

Indeed, the manuscript is focused on the later developmental stages, when the PGC migration ceases. Therefore, we compare the speed of cells at these developmental stages (from 14 hpf onward) to those reported for earlier stages (~2 $\mu\text{m}/\text{min}$ at 7-9 hpf; see Reichman-Fried et al., 2004). The latter (PGC speed measurements at 7-9 hpf) were reproduced in this work- please, see Fig. 2F and Fig. 2I, where the speed of endogenous (host) PGCs was measured at 8 hpf.

Reviewer 3: SUMMARY OF THE ADVANCE MADE IN THIS PAPER AND ITS POTENTIAL SIGNIFICANCE TO THE FIELD

This is an interesting paper investigating the mechanism of how cells stop migration during development. Previous literature has shown that the dynamic responses of receptors to external ligands are important in cell stopping. Here the authors introduce an intrinsically programmed mechanism to stop cells based on changes of expression of an RNA-binding protein, which in turn regulates migration-associated genes. These findings have broad relevance, as they may suggest mechanisms by which other cell types seize migration in various contexts, for example in disease pathologies, such as cancer metastasis. The experimental set includes genetic manipulations, transplantations and live imaging and generally supports the claims of the authors. The paper also includes elegant approaches such as RNA tomography and tissue-specific gene silencing to obtain spatial maps of gene expression and to interrogate relevant mechanisms. The data presentation and writing are of high quality, and I only have minor comments and suggestions for improvement.

SUGGESTIONS TO AUTHORS

Minor points:

-In the gene silencing section, Fig. S1C and D, it would be helpful to have a quantitative estimation of the level of knockdown, to indicate the levels achievable and relate to phenotypic changes observed in S1E and S1F.

We thank the referee for raising this important point, which was previously missing from the paper and is now incorporated into the new version of Figure S2, panel E. The quantification of the PCT ablation was performed in two different ways: 1. Counting the remaining PCT cells and 2. Measuring fluorescence intensities of the PCT marker.

-RNA tomography is mentioned at an early stage in the manuscript, but we don't see relevant data before figure S4. Although there is a table with data, it would be good to have a diagram of the assay and charts of the relevant datasets.

According to the referee's suggestion, we include a new figure, Figure S1, explaining schematically the RNA tomography (TomoSeq) experiments we conducted (Fig. S1A) and present a pie-diagram with representative in situ images (Fig. S1B) of the dataset results listed in the Table S1.

-The transplantation experiments are important to distinguish cell autonomous from non-cell autonomous effects. The controls included in figure S3 may be helpful to include in the same main figure, to facilitate interpretation. In addition, can the authors comment on the estimated purity of the transplants in PGCs. Is there a significant proportion of non-PGC cells carried over that may contribute to phenotypes?

As suggested by the referee, the transplantation control panels are now included in the new version of the Fig. 2.

As noted by the referee, employing this specific method, a mixture of PGCs and somatic cells was transplanted. The size of transplants was kept comparable in all experiments. We therefore reasoned that the number of somatic cells transplanted was similar for all treatments. The transplanted cells spread within developing hosts due to gastrulation and are outnumbered by the somatic cells of the host. We therefore consider the co-transplanted somatic cells (ca 50 cells as compared with 3000-4000 host cells) insignificant to the observed differences in the PGC migration speed.

-The actin/Ezrin distribution is suggested to be altered during PGC development, but this would need some quantification.

This is an important point and we have included polarity quantification in the new version of Figure 3 (Panels C and E).

-In the cartoon diagram of Figs 3F and S4 can the authors use consistent references to 'RNA-seq' versus 'NGS'.

We have changed Figures 3F and S4 accordingly and refer to it as RNA Seq.

-In Fig. S4, the authors report transcriptome data from Cxcr4b knockout fish but the rationale of the use of this strain is not clearly explained in the results.

The reason for conducting RNA seq also on cells from Cxcr4b knockout embryos in addition to control embryos was to determine if the changes we observe result from extrinsic signals, or reflect a cell-autonomous event. To clarify this point, we rephrased the sentence in the Results section as follows: "RNA-seq analysis was performed for PGCs at the gonad region (Fig. 3F) and for ectopically-located germ cells (in embryos lacking functional guidance receptor Cxcr4b, Fig. S4A)."

-In this sentence, can the authors give the names of all the RNAs injected "we injected embryos with a combination of RNAs encoding for germ plasm factors, including Dnd1, to transform all the cells in the embryo into germ cells, hereafter referred to as "induced PGCs" (iPGCs) (Wang et al., 2023)."

As requested, we have included an additional section in the new version of Supplementary Information that explains the PGC induction method in detail ("PGC induction" Section).

-Can the authors more clearly comment on a possible relevance for the observation that *dnd*-high cells have lower germ-line transmission? Could this suggest that there is a functional/developmental trade off with PGC migration? It would be good to discuss this point along with other references where such trade-offs might have been observed.

*We believe that there is a tradeoff between the necessity of a PGC to reach the target region and the gradual loss of motility required for interactions with the somatic cells at the region of the forming gonad. Thus, germ cells with high *Dnd1* levels retain high motility/contractility levels, which is manifested in elevated displacement. At the same time, it impairs their ability to form stable connections with gonadal somatic cells. In other words, we think that as compared with endogenous cells, the manipulated, more motile germ cells are less efficient in integrating into the gonad and are therefore outcompeted. As a result of such a competition, we observe a reduced level of germline transmission of PGCs with high *Dnd1* levels.*

*We have added the following statement to emphasize this point: “We hypothesize that the reduced germline transmission rate of *Dnd1*-high PGCs can be attributed to their elevated contractility and limited capacity to form functional interactions with the somatic part of the developing gonad. An additional factor that could link PGC displacement with the ability to generate gametes is the fact that the position of PGCs along the migratory route could elicit changes in pluripotency, and epigenetic reprogramming (Jaszczak et al., 2025).”*

-The authors could give some more background into the biological function of *dnd*, to better appreciate the biological mechanisms and cite any other studies where similar regulation of migration might have been observed.

*Indeed, the background to the biological function of *Dnd1* is given in a condensed form, with a primary focus on its role in germ cell motility, which is the main theme of this work. We now include additional references on the mechanisms of *Dnd1* function and describe it as - “One way to explain the drop in cell motility is the dramatic decrease in the expression level of *dnd1* during the first 35 hours of embryonic development. Indeed, we and others have previously found that *Dnd1* is essential for germ cell motility and fate (Goudarzi et al., 2012; Gross-Thebing et al., 2017; Mall et al., 2021; Ruthig et al., 2019; Wang et al., 2025; Weidinger et al., 2003; Westerich et al., 2023; Youngren et al., 2005). The mechanism by which *Dnd1* exerts its function involves binding target RNAs, stabilizing them and enhancing their translation (Aguero et al., 2017; Kedde et al., 2007; Ruthig et al., 2023; Zhang et al., 2021).”*

-In the introduction, it would be helpful to give more background on alternative mechanisms of stopping, for example chemoattractant receptor desensitization (e.g. Minina et al. 2007, Coombs et al. 2019, Kienle et al. 2020), and differential adhesion (Yamaguchi et al. 2022). This will help appreciate better the novelty of this paper regarding cell-intrinsic mechanisms of stopping.

We have included the above-mentioned mechanisms in the Introduction of the revised manuscript as - “Several mechanisms regulating motility in single migrating cells have been described; for example, desensitization of the chemoattractant receptor (Coombs et al., 2019; Kienle et al., 2021; Minina et al., 2007) and differential adhesion ((Cortés et al., 2003), reviewed in (Miskolci et al., 2021; Yamaguchi and Knaut, 2022)).”

-Another suggestion for the last paragraph of the paper is to discuss the broader relevance of the findings, in other cell types and biological contexts.

*Following the Reviewer’s suggestion, we have added the following sentences to the revised version of the manuscript to place mRNA decay in the broader biological context: “Analogous to our findings on the role of the RNA-binding protein *Dnd1* in zebrafish, the regulation of RNA stability, localization, and translation controls a wide variety of cellular processes in other organisms. For example, the function of RNA-binding proteins that control RNA decay rates is critical for regulating signaling cascades in neurogenesis as well as for the fidelity of neuronal cell migration (La Fata et al., 2014; Messaoudi et al., 2024; Zhang et al., 2025).”*

Second decision letter

MS ID#: dev.205271R1

MS TITLE: Cell-autonomous control coupled with tissue context regulates the cessation of migration at the site of organ development

AUTHORS: Katsiaryna Tarbashevich; Zahra Labbaf; Jan Schick; Lucas Kühl; Sargon Gross-Thebing; Moritz Ophaus; Michal Reichman-Fried; Dennis Hoffmann; Martin Stehling; Jochen Seggewiss; Jan Philipp Junker; Christian Ruckert; Johanna B. Kroll; Erez Raz

ARTICLE TYPE: Research Report

Dear Dr Erez,

I am delighted to tell you that your manuscript has been accepted for publication in Development, pending our standard publication integrity checks.

Reviewer 2

In this resubmission, the authors responded to reviewers' critiques and suggestions with additional information in the text and several additional quantifications. Namely, speed was measured following PCT ablation and when *Deadend1* is increased. They also quantified actin/Ezerin polarity and assessed PCT ablation efficiency. These changes and additions clarified and strengthened the rigor of this very interesting and well-done study.

The additional quantifications revealed a more modest increase in PGC speed under high *Dnd1* conditions relative to the increase in PGC displacement (Fig 4B and C). In their response, the authors state that displacement is also more affected than speed when comparing early vs late PGCs. As a possible explanation for this observation, the authors added the following text "We attribute the more pronounced differences in PGC displacement compared to the differences in speed to the fact that cells possessing higher contractility can better cope with obstacles in the environment without altering their migration direction as much. This allows PGCs to migrate further, which is manifested in the increase in displacement, while speed is less affected (Fig. 1, Fig. 4B, C)." Given this added statement, the authors may want to add a comment comparing the impact of increased *Dnd1* (Fig 4B,C) with *Cxcr4b* knockdown (Supp Fig S3).

Minor point: The legend for Fig 4D mentions error bars but these error bars are not shown in the graph.

Reviewer 3

The authors have performed a thorough revision and addressed my comments. I have no further concerns and recommend this nice study for publication.